# Institutional and Legal Framework of the Brazilian Energy Market: Biomass as a Sustainable Alternative for Brazilian Agribusiness

**Marcia Carla Pereira Ribeiro** [1,2,*], **Caroline Paglia Nadal** [2], **Weimar Freire da Rocha Junior** [3], **Rui Manuel de Sousa Fragoso** [4] 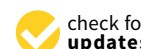 **and Cleber Antonio Lindino** [5]

1    Law School, Pontifical Catholic University of Parana, Curitiba 80215-901, Paraná, Brazil
2    Department of Private Law, Federal University of Paraná, Curitiba 80020-300, Paraná, Brazil;
     carol.paglia@gmail.com
3    Regional Development and Agribusiness, State University of Western Paraná,
     Toledo 85900-001, Paraná, Brazil; wrochajr2000@gmail.com
4    Business Management Department, University of Évora, 7000-645 Évora, Portugal; rfragoso@uevora.pt
5    Environmental Science, State University of Western Paraná, Toledo 85900-001, Paraná, Brazil;
     lindino99@gmail.com
*    Correspondence: marcia.ribeiro@pucpr.br

**Abstract:** The present article discusses the institutional model adopted in Brazil regarding energy production through biomass. The utilization of renewable energy and clean sources of energy is a characteristic of Brazil's energy matrix. Recently, the production of energy through biogas—and biomass in general—started to gain force. The concessionaires of energy, as well, started to discipline its utilization and selling, creating an impediment to the commercialization of energy produced by micro or mini generation outside the free energy market, as well as a prohibition of binding the contracts to the value of electric energy. Even though, it is possible to create a network of contracts that interconnect the producers and the consumers. In this line, the theoretical-empirical method was used to conclude that the model's difficulties—and specially the legal limitations—can be overcome by adopting a network of contracts capable of subjecting renewable energy generation to an energy compensation model.

**Keywords:** biogas; Brazilian agribusiness; business model; energy

## 1. Introduction

Thinking about sustainability in the field of generation and consumption electricity requires an initial reflection about the most used sources for its production.

The global primary energy consumption amounted to 13,864.4 million tons oil equivalent (TOE) in 2018, an increase of 2.9% over 2017. Of this total, renewable energy (except hydro) corresponded only to 4.05% (561.3 million TOE). In Brazil, the primary energy consumption amounted to 297.6 TOE, in which 7.93% (23.6 TOE) were originally from renewable energy, including biomass [1].

In the Brazilian energy matrix, in 2018, the bioenergies corresponded to 31.1% of the total supply. The internal capacity for electricity generation from biomass corresponded to 14,790 GW (9% of the total) [2] (p. 4).

The choice of the energy matrix also relates to the system costs and regional conditions. For agroindustrial regions, biomass can be a viable raw material for the production of clean and renewable energy and, at the same time, a form to minimize the environmental impacts of agroindustrial production.

After presenting the concept of biomass, this article shows the Brazilian's legislative panorama regarding the micro and mini distributed generation as a form of energetic production that has some benefits when working as a policy to stimulate the generation of renewable energy.

In its final part, it demonstrates the need to organize a contract network and to build a solid model of distributed generation for sharing purposes, through the compensation of the consumption tariff for the energy produced.

## 2. Biomass

According to the US Energy Information Administration, "biomass is organic material that comes from plants and animals, and it is a renewable source of energy. Biomass contains stored energy from the sun. Plants absorb the sun's energy in a process called photosynthesis. When to the burning of biomass, the chemical energy in biomass releases heat. Biomass can be burned directly or converted to liquid biofuels or biogas that can be burned as fuels" [3].

Food, yard, wood waste in garbage, animal manure and human sewage are substrates that can be converted to biogas, which can be burned as a fuel or generate electricity in power plants [3].

Agroindustry waste generation in Brazil represents a huge amount of initially underused material that can serve as raw material for power generation. Forster et al. (2013) [4] determined the availability of agricultural residues, animal waste, and its respective analyses as to generation potential [5]. They considered the presence of those biomasses in the sites and around them, as well as production systems in use according to the agronomical needs and weather characteristics of different Brazilian regions. The calculus of the generating potential (GP) index has been done by the relationship between the generated total residues by type and physical production (t/ha/t/ha). The main selected products and the GP index are presented in Table 1.

**Table 1.** Estimates of generating potential index (GP) for agricultural residues and animal waste in Brazil.

| Feedstock | GP (t./culture) |
|:---:|:---:|
| *Agriculture residues* | |
| Sugarcane | 0.22 |
| Soybean | 2.05 |
| Maize | 1.42 |
| Rice | 1.49 |
| Cotton (perennial) | 2.95 |
| Orange | 0.50 |
| Wheat | 0.20 |
| Cassava | 0.20 |
| *Animal waste* | |
| Poultry (chicken) | 1.58 |
| Swine (pork) | 0.06 |
| Bovine (beef) | 0.07 |

Source: FORSTER et al. (2013, p. 77) [4].

Table 1 shows that agriculture residues such as soybean, cotton, and maize, has great potential to generate residues, and by consequence more potential to supply energy. On the other hand, in animal waste distinction poultry has great potential than others. Data exposed, v.g., each 1000 tons of soybean produced will generate 2.05 residues as stem, leaf, straw, dry string bean, etc. in land; cotton generates 2.95 tons. The producers have to lead with and assure a correct destination. Using part of these residues to produce energy could assure that both the best destination follows parameters of sustainability and offers a resource of energy.

The production of biogas involves several benefits, such as the generation of energy from renewable sources, the environmental preservation due to the reduction of waste or waste in natural

resources, and the reduction of greenhouse gas emissions (GGE), besides driving the local economy and generating jobs.

The production of biogas from agroindustrial waste mainly involves anaerobic processes, in which organic matter is principally converted to methane gas ($CH_4$), carbon dioxide ($CO_2$), water, and other gases in smaller proportions ($H_2S$, $H_2$, $NH_3$) through a consortium of bacteria in a complex reactions process, inside a reactor called biodigester. After the appropriate residence time of the waste or manure in the digester, it may be stored in stabilization ponds, and then applied as a biofertilizer in crops according to the agronomic specialist recommendation [6].

Several factors influence the anaerobic digestion process and therefore the efficiency of biogas production. These include temperature, pH, total alkalinity, the presence of macro and micronutrients, the type of residue inserted, and the carbon-nitrogen ratio (C/N ratio between 15 and 30). However, it is important to emphasize the importance of correct management of the digestion system by ensuring adequate residence time, to avoid crusting and preferred residue pathways, the excess of water, contamination with detergent or disinfectant from washing the breeding sites, and presence of toxic heavy metals.

## 2.1. Biogas in the Brazilian Concept

Historically, the first biodigester in Brazil was installed in the official residence of the Presidency of the Republic in November 1979 and followed the Chinese model. That was the first phase of expansion of the use of biodigesters between 1980 and 1984, with around 3000 biodigesters installed at the end of the period. Operational problems led many farmers towards abandoning the technology in the following years. The second phase of expansion took place only in the 1990s, impacted by the Rio-92 Environment Conference, by the mitigation of the greenhouse effect, and the use of carbon credits. In 2015, biogas production was 1.6 million cubic meters per day.

Recently, the use of biodigesters for biogas production has the environmental appeal associated with renewable energy generation, from the use of stationary motors for electric power generation. The current generation potential of biomethane is 14.3 million $m^3$ per day and in terms of electricity, generation is 3478 MW from agricultural waste. With the use of municipal solid waste, the potential use of biomethane is 4.2 million $m^3$ per day and electricity generation is 868 MW. In addition to these, there is the possibility of using sludge from sewage treatment plants and other industrial waste. Energy biogas applications in Brazil are thermal (49%), electrical (44%), mechanical (5%), and biomethane (2%) [7].

Figure 1 shows the flowchart of the main technological trajectories of biogas in Brazil.

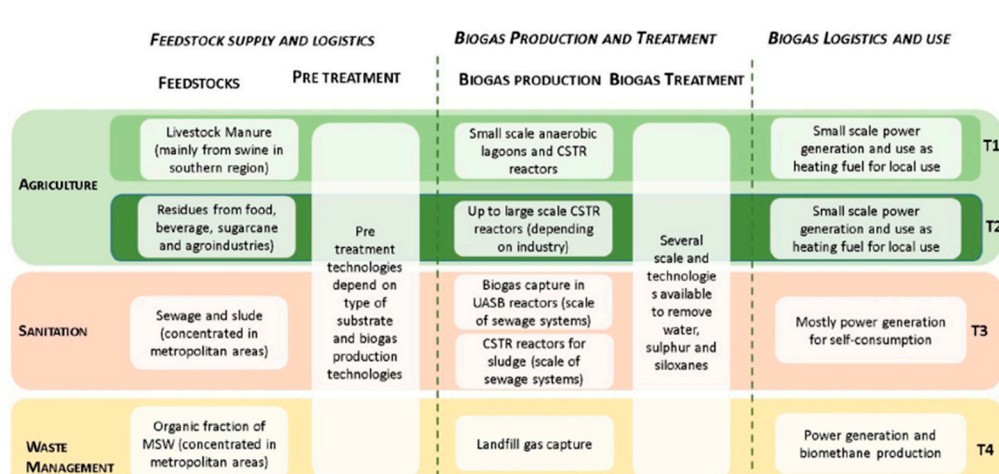

**Figure 1.** Main technological trajectories of biogas production in Brazil [7].

The Brazilian Constitution (1988) has several articles dealing with the environment, including a ministry and an autarchy that regulates and oversees environmental activities regarding environmental preservation. The chapter on environment contains article 225 that specifies that item one paragraphs IV and V and item two are part of the constitution that has a direct relation to production of biogas.

The constitution is a set of rules of institutional environment that fundaments legal ground rules that establishes the basis for production, exchange, and distributions. Those rules govern all economy activities and constrain the action de economic agent [8]. That is why the energy production in Brazil is submitted, at once, to the legal constitutional bases and to standards made by administrative authorities.

## 2.2. The Problem

The adoption of renewable energy sources, as instruments compatible with the search for environmental sustainability has its main obstacles in the restrictive regulation imposed on the activity of generation and distribution of energy, as well as in the values of investments that will be associated with the implementation of power plants.

In particular, in the field of energy production from biogas, barriers will only be overcome by sensitizing the agents operating in agroindustry as to the role that they can play in improving and preserving the environment, accompanied by an integrative action that, through the union of efforts and investment capital, makes this type of energy production economically viable.

## 3. Electric Energy Production

Following the notes on biomass and the understanding of the Brazilian landscape regarding the generation of renewable energy, some concepts related to energy generation and the institutional environment characteristic of the sector will be presented.

## 3.1. Micro and Mini Generation: Definition of the Regulatory Authority

In Brazil, since 17 April, 2012, with the publication of Resolution 482/2012 of the National Electric Energy Agency—ANEEL, Brazilian consumers may install small generators for the purpose of generating and compensating electricity—as long as its within the same distribution network.

The initiative resulted from the promotion of a more sustainable generation of electricity, with greater capillarization of energy sources and use of natural resources. Thus, it was possible to generate power closer to the consumption centers. The aim was to promote the production of energy through renewable sources—hydro, wind, and photovoltaic generation—and cogeneration, such as biomass and solar thermal.

Among the benefits of shared generation are: The postponement of future extensions of transmission, distribution, and power generation networks; the diversification of the country's energy matrix; the low environmental impact; reduction in network loading and minimization of losses [7] (p. 7).

The counterpoints, however, are the increase in small generators scattered throughout the distribution network; the increased complexity of network management; the difficulty in charging the system for the use of the electricity system; the form of taxation and the constant need for change, by the distributors, of the operation, control, and protection procedures of the lines [7] (p. 7).

In order to promote the development of distributed generation activity, the government has authorized a series of tax mitigations. As an example, the ICMS—a state tax levied on the circulation of goods and services—was reduced, through the National Council of Finance Policy (CONFAZ) normative n. 6/2015, which granted exemption in the operations related to the circulation of electricity on the system of compensation of energy. Moreover, it determined the incidence of PIS and COFINS—taxes used to fund what is called in Brazil the "social security"—only on the positive difference between the energy consumed and the energy injected by the unit, instead of comprehending all generated energy [9].

However, this type of power generation cannot be confused with the sale of electricity. Furthermore, national legislation prohibits those who participate in the free market of electricity and practice the commercialization of energy to register as small generators. Thus, micro and mini generation regulations only apply to participants in the captive electricity market.

The legislation also brought requirements regarding the permitted generation modalities. Thus, the generation of electricity by consumers themselves must use qualified cogeneration or renewable sources of electricity. Qualified cogeneration means the process that results in the combined production of heat and mechanical utilities, which are generally wholly or partially converted to electrical energy [10]. Although renewable sources include hydroelectric generation, only wind, photovoltaic, and biomass generation fall within the modalities allowed by Decree 5.163/04 for generation purposes by consumers themselves.

Still, power generation was divided by ANEEL between micro and mini generation. The first comprises a power plant with an installed capacity of 75 kW or less and the second, a production potential greater than 75 Kw and less than 5 MW.

Under the clearing system, the energy generated by consumers is provided through a free loan to the utility. In exchange, it is offset in the future energy bill, with the consumption of active electricity [11]. If the generation exceeds the energy consumption, the distributor shall reduce the excess consumption in subsequent months [12]. The electric power credit will be valid for 60 months [13].

However, consumers may, in addition to compensating with their own expenses, direct the electricity generated to other consumer units, provided that they are previously registered under some of the sharing models authorized by ANEEL.

### 3.2. Allowed Systems in the Brazilian Law

To enable energy sharing of the surplus production with other consumer units, some modalities of electric power sharing were established by ANEEL.

Currently, national legislation permits the existence of three systems: (i) Multiple consumer units; (ii) shared management venture; and (iii) remote self-consumption project [14].

The multiple consumer units are characterized by the use of electricity independently. In this model, a consumer unit generates energy for several other smaller units, which constitute a fraction with personal use. This system requires all units to be in neighboring properties, being forbidden the use of public roads, air or underground passages, and third-party properties that are not part of the project.

A concrete example of this type of venture is the condominium. Usually, in this type of organization, in common areas of residential condominiums, power plants may be installed. The energy generated is distributed among all consumer units, which, in this case, are the condominium houses. This setting may be extended to borderline properties, provided that the utility grid is not used for the distribution of electricity.

In the opposite, the shared management venture is the gathering of consumers within the same concession area with facilities directly connected to the distribution system. Thus, it is possible that the place of generation of electricity is distant from the property of those who will consume it. The connection between the generator and the consumer is through the utility facilities.

The organization of this type of enterprise is usually through the union of consumers in a consortium or cooperative, so that the percentages of energy directed to each consortium or cooperative are established in advance.

Finally, remote self-consumption is characterized by the existence of several consumer units belonging to the same holder, be it a legal entity or an individual. Therefore, the energy produced in a given generating unit can be harnessed by it in all its properties.

In order to put into practice the models allowed by regulatory authorities, business law is the branch of law that offers the models that will enable the legal structuring of business.

### 3.3. Restrictions

Electricity generation in the model analyzed does not allow the sale of the surplus, either to the concessionaire itself or to other consumers. The sale of electricity, in fact, is regulated by the Normative Resolution 678/2015, which does not deal with the compensation of energy.

In the sharing model, it is not possible to carry out operations in which the consideration in monetary value (expressed in currency) and the electricity itself are identified. Exactly in this sense, RN 482/12 ANEEL states that "the distributor may not include consumers in the electric energy compensation system in cases where it is detected, in the document proving the ownership or ownership of the property where the micro-generation is installed or distributed generation, that the consumer has rented or leased land, lots and properties under conditions in which the value of the rent or lease is in reais per unit of electricity".

This forecast was included in the legal text due to the commercial practice of selling electricity produced in certain plants, for example. Commercial practice revealed that generating units were selling energy by adopting a calculation between the amount of money to be paid and the energy directed to the consumer that compensated it.

However, there is no legal prohibition that the parties involved in a mini or micro distributed generation project (owner of the generation equipment, owner of the installation site, users of surplus generated for compensation purposes) are bound by an energy management that establishes a form of remuneration, provided that it is not linked to the value of the energy produced.

### 4. Energy Management

The generation of energy through biomass is a possibility aligned with the search for environmentally appropriate options regarding the form of generation (clean energy), as well as acting as an alternative for the disposal of tailings associated with activities that characterize the agribusiness.

However, although desirable, micro and mini generation meet limits established by Brazilian law: The commercialization of surpluses is forbidden.

Therefore, in order for the desirable alternative to become a more frequent option for agribusiness agents that consume significant amounts of electricity, it is essential that the necessary investments for the installation of micro or mini power generation through biomass can be allocated.

This means that all those directly interested in the process (producers who need to dispose of waste arising from, for example, animal husbandry; electricity consumers who will have better results if they can reduce the cost of energy; holders of investment power or technology of energy-generating motors; owners of areas that can lease their properties for the installation of equipment) need to overcome the barriers through the elaboration of a network of contracts capable of making the interests compatible and the barriers to the implementation of the system feasible. The surplus energy sharing scheme is generated.

### 4.1. Contracts Network

Considering governance structures as adopted by Williamson [15], the user of electricity can purchase it from the free market (market), can link it to a concessionaire using a contract, or choose to internalize its production (firm). In the latter case, when it comes to renewable energy from biomass, the major difference is that it will also be given a destination to the waste inherent to agroindustrial activity, a legal imposition aimed at ensuring environmental sustainability. Therefore, in addition to the potential benefit in terms of operating costs, the rural producer will assign a destination to the tailings that may even result in the production of fertilizer to be used in cultivation.

The cost of electricity produced predominantly in Brazil tends to have a decisive impact on the overall costs of production activity, being predominantly derived from the hydroelectric matrix [16] (p. 37).

The cost becomes even more significant when weather conditions are unfavorable, which makes the generation complemented by thermoelectric plants [16] (p. 23).

For that, the scenario in which agribusiness producers are subjected to the cost of energy, coupled with the importance of agribusiness to the country's economy, reinforces the importance of finding cheaper and environmentally friendly energy sources.

For the production of energy through biogas generation to become viable in Brazil and in order to overcome the difficulties arising from the legal discipline of the sector, the adoption of a contract network proves to be the most appropriate way.

### 4.1.1. Place

Under the Brazilian regime, the production of electricity for commercialization is subject to authorization by the Government, through concession agreements, which allows a private agent (with or without public participation) to act in order to produce energy. The grant attributes the character of a public service concessionaire to the beneficiary company.

In the state of Paraná, the largest concessionary company is COPEL—Companhia Paranaense de Energia. It is required for the authorization of power generation for compensation purposes that the user registered in COPEL is the owner or lessee of the property in which the energy will be produced. The energy compensation regime assumes that the generating unit and the units that will compensate it all buy energy from the same utility company.

Thus, in the case of a livestock breeder (LB), for example, who is willing to use the tailings from their production or concentrate the production of other producers' waste (OPW), but to do so will associate with other investors (IN) or owner of the biodigestion engines (BEN), it being in the interest of the parties that the energy management is carried out by the owner of the engines (BEN), a rural property lease may be entered into, which will appear as the LB lessor and as renter BEN.

In the event that fractionation is not possible, the property will be fully leased. In this case, in order for the LB to continue in the exercise of its activities, a right of use contract may be signed for the area that exceeds that used in micro generation (machinery and tailings tank), in which the assigned will be BEN and LB the assignee.

### 4.1.2. Tailings Suppliers

In order to optimize the use of equipment and the consequent production of compensable energy, it is possible that the LB and BEN join OPW who are interested only in giving proper disposal to the waste—which is a legal imposition of environmental preservation—or also interested in participating of the sharing of energy produced.

Regarding the receipt of tailings for biogas production, there is the option of building an underground network through which the tailings can be moved from their place of origin to the reservoir, or the use of another form of transport.

The contract should include, among other questions, who will be responsible for transportation costs, periodicity, whether there will be any form of remuneration from the LB or BEN.

### 4.1.3. Users

In the proposed system, Brazilian allegation admits that energy production is shared but not traded.

There is the possibility of self-consumption, in which case there will be no need for any other contract except with the energy concessionaire, but the condominium, cooperative, and consortium system among several users is also allowed, for the purpose of offsetting the energy produced by the BEN.

The condominium system can only be used to share the energy produced by limiting consumer units. If this is the case, a contract should be formalized covering the consumer units, what percentage will each use of the production be compensated. The contract should also indicate the minimum advance deadlines for percentage change or consumer unit change covered by the contract.

In the cooperative system, those interested in offsetting come together through a contract also for the purpose of sharing via offsetting. Due to the structure chosen, there will be a need for the specific rules applicable to cooperatives to be respected. In Brazil, legal discipline is found in Law 5.764 / 71 and the Brazilian Civil Code. Among them, there is the need for at least twenty partners for its constitution, as well as the submission of the members to a subsidiary liability regime for the company's debts, which may be limited or unlimited. When analyzing the available structures, it should be taken into account that the replacement of a member of the cooperative will depend on a change in the articles of association, which may take a certain amount of time and will be done only after the change that the percentages allocated to the member who leaves the company—may be directed to compensation to another user/cooperative.

Finally, association via a consortium contract may also be considered. The consortium here appears as a system that allows users of electricity to adhere, through autonomous contracts, to a contract signed between the consortium and the electricity concessionaire, which establishes the rules for compensation of the energy produced. The consortium, by operating through autonomous contracts, allows greater agility in the substitution of users, and does not include the express norms of responsibility for obligations of the consortium, because the consortium provided for energy sharing purposes does not form a person, that is, it does not lead to the creation of a subject of law.

### 4.1.4. Contract with the Concessionaire Company

As already indicated in the article, in the Brazilian system the production of electric energy and its distribution are considered services of public authority. Therefore, for the adoption of a shared micro or mini power generation system, it is mandatory by law that the consuming units, from the place of the energy generation, as well as the units that will use the compensation of the energy produced by the consumed energy, are linked to the same concessionaire.

Therefore, the adoption of any of the models allowed by law for the operation of the system will depend on a deal with the concessionaire company, whose approval and consequent start of operations will be subject to the concessionaire's scrutiny, which will evaluate the compliance with the legal requirements, regulations (issued by the federal regulatory agency, ANEEL—National Agency of Electric Energy), as well as any internal legal and operational guidelines.

Constraints related to the deadlines for changing the terms of the covenant on percentages of energy produced to be offset, as well as the definition of consumer units benefiting from offsetting, should be observed, otherwise the system efficiency will be lost, which could lead to a potential increase of transaction costs of consumer units.

Under national legislation, the percentages produced for the previously defined consumer units cannot be used by other units of the system, and are therefore linked to the unit to which they were contractually assigned. Moreover, the replacement of a consumer unit, which is withdrawn from the system, is subject to the rule that any compensation that has not been operated will remain available to the unit that withdraws for a period of five years and cannot be destined for other units of consumption.

### 4.2. Management for Efficiency —Need to Communicate Ownership Change in Advance and Percentage Adjustment when Required

The purpose of sharing micro or mini generation by offsetting the energy produced by the energy consumed presupposes initial investments that can be considered impeditive from the individual perspective of a rural producer. On the other hand, the use of the compensation benefit, can only have its benefits maximized if submitted to specific management that can be consolidated by drawing up an energy management contract.

An energy management contract, the content of which shall be drawn up in the best interests of the parties to the contract and which does not prevent the definition of a—necessary—remuneration commensurate with the size of the operations associated with it. The law prohibits compensation to be linked to the value of energy produced or offset.

Regarding investment management, in which several stakeholders will join efforts and capital for the implementation of the system, the costs involved will be associated with costs for the approximation of potential stakeholders (energy producing unit, consuming units, interested in taking advantage of the tailings disposal system, owners of areas compatible with the purpose of power generation, energy concessionaire company, owners of biodigestion equipment and other necessary equipment, construction works), and drafting of the necessary contracts. The energy management agreement may include such actions.

The management contract itself will establish the conditions for changing the terms agreed by the parties before the concessionaire, but should also consider, through data analysis, a plan capable of leading the compensation system to maximize its results.

For example, depending on the line of business performed by the sharing companies, there may be periods of greater or lesser intensity in the use of electricity, which should lead the manager to a plan that allows the percentages allocated to the participants to be targeted so that the potential surplus of one consumer unit is transferred over time to another unit, avoiding the loss of the opportunity for immediate and guaranteed offsetting of compensable energy.

The energy manager should also provide for the planned replacement of users tied to the systems in a manner that avoids remnants of compensable amounts not usable by the other units participating in the sharing system.

It is believed that the legal impediment in Brazil of the commercialization of energy produced by micro or mini generation outside the free energy market, as well as the prohibition of binding the contracts to the value of electric energy, do not represent an impediment to the contract of energy management prepared by stakeholders.

In fact, it is possible to create contracts using the management of energy and even rent of units of generators in order to exchange the energy produced with biomass with individual consumption. The legal impediment is only related to the direct relation between energy and the value required, but many other forms of contract can be constructed in line with this legislation.

That is the reason why the use of the commented contract network attends to all legal conditions of Brazilian's law and electric system, allowing to implement energy generation systems through biomass.

## 5. Conclusions

Energy produced from biomass is an alternative compatible with the goals of sustainability and cost reduction in Brazilian agribusiness.

The agribusiness importance for the Brazilian economy, the need to attribute a correct destination to the produced waste, and the incentive to adopt renewable energy matrices contrast with the legal restrictions on the negotiation of surplus energy produced by micro and mini producers. The costs to install the biomass generator, however, prevent its use by a large part of small Brazilian rural producers.

In order to become viable, given the characteristics of the national rural economy and the country's institutional environment, it can be necessary to adopt a network of contracts able to bring players interested in the system together, to guarantee a form of management that maximizes the potential of energy compensation.

It is believed that institutional Brazilian problems can be repeated in other developing economies and that the present article can assist in searching for local compatible solutions with sustainability ideals in energy production.

**Author Contributions:** Formal analysis, R.M.d.S.F.; Investigation, M.C.P.R., C.P.N., W.F.d.R.J. and C.A.L. All authors have read and agree to the published version of the manuscript.

**Funding:** This research was funded by Fundação Araucária de Apoio ao Desenvolvimento Científico e Tecnológico do Estado do Paraná and Pontifícia Universidade Católica do Paraná, TC 030/2019.

**Conflicts of Interest:** The authors declare no conflict of interest.

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
