# Peer review of "Institutional and Legal Framework of the Brazilian Energy Market: Biomass as a Sustainable Alternative for Brazilian Agribusiness"

_sustainability, doi:10.3390/su12041554_

Round 1

Reviewer 1 Report

My major problem with article is the loose connection between the title and the text. According to the title an analysis of biomass based renewable production is expected, but the text is about the institutional and legal framework of the Brazilian energy market. In this content, it would be interesting to make e.g. a comparison with other systems/countries. I would like to read more about the RE expansion related to this background, the positive effects on the different stakeholders (e.g. in a form of a PEST or PESTEL analysis) or sustainablity issues in order to make the article more balanced.

I miss the source, as well as some analysis, for table 1. I would advise the authors to compose a diagram on the different hubs to make them easier to understand.

Author Response

Dear revisor,

Here is the revised document!

Reviewer 2 Report

The paper is very interesting. It contains important issues and useful in explaining the concept of b Brazilian’s legislative regarding the micro and mini distributed generation as a form of energetic production that has some benefits when working as a policy to stimulate the generation of renewable energy.

However the adoption of renewable energy sources, compatible with the search for environmental sustainability has its obstacles in the restrictive regulation imposed on the activity of generation and distribution of energy in micro and mini energy generation meet limits established by Brazilian law but the commercialization of surpluses is forbidden.

Similar problems occur not only in Brazil, but concern the global sphere in the production of sustainable use of available energy sources, in particular the possibilities of processing waste products and renewable energy.

The work is a review, but is based mainly on the provisions of current Brazilian law and reports as well as the latest articles. The current situation was assessed and some solutions were proposed.

I suggest Authors should expand the content of the abstract, add more information. The text is understandable and correct in terms of style; It only requires minor grammatical changes, especially in References, where month names are spelled incorrectly.

Author Response

(The authors gave the same response as above.)

Reviewer 3 Report

A well presented and interesting article. Some minor editorial points to be corrected. I would have liked the authors to present their views on how the future contracts for the micro/mini generation of electricity should be constructed. They would appear to indicate that there is no legal impediment so hence what should these contracts look like. I would also suggest that the authors reconsidered the working of the final paragraph in Section 4.2 as it is a little unclear as to their opinion on the impediment to the stakeholders - the paragraph suggests that there is not an impediment but it was unclear if this was the opinion of all the stakeholders or just the legal point. The opinions of the micro/mini generator businesses with regard to their views on the business opportunities or barriers would be useful if appropriate. 

Author Response

(The authors gave the same response as above.)

Round 2

Reviewer 1 Report

I am fine with the revised version.